# Reliable Sea Water Ro Operation with High Water Recovery and No-Chlorine/No-Sbs Dosing in Arabian Gulf, Saudi Arabia

**DOI:** 10.3390/membranes11020141

**Published:** 2021-02-17

**Authors:** Hiroki Miyakawa, Mohammed Maghram Al Shaiae, Troy N. Green, Yohito Ito, Yuichi Sugawara, Makoto Onishi, Yoshinari Fusaoka, Mohammed Farooque Ayumantakath, Ahmed Saleh Al Amoudi

**Affiliations:** 1Hitachi, Ltd., 5-1, Soto-Kanda 1-Chome, Chiyoda-ku, Tokyo 101-0021, Japan; makoto.onishi.hv@hitachi.com; 2Saline Water Conversion Corporation, Desalination Technologies Research Institute, Al-Jubail 31951, Saudi Arabia; MAl-Shaiae@swcc.gov.sa (M.M.A.S.); TGreen@swcc.gov.sa (T.N.G.); MAyumantakath@swcc.gov.sa (M.F.A.); AAl-amoudi@swcc.gov.sa (A.S.A.A.); 3Toray Industries, Inc., 1-1, Sonoyama 1-chome, Otsu, Shiga 520-8558, Japan; yohito.ito.a2@mail.toray (Y.I.); yuichi.sugawara.w9@mail.toray (Y.S.); yoshinari.fusaoka.g3@mail.toray (Y.F.)

**Keywords:** reverse osmosis, high water recovery system, reliable operation, no-chlorine/no-sbs dosing, biofouling, membrane biofilm formation rate

## Abstract

For providing advanced desalination the combination of the improvement of water recovery ratio in the reverse osmosis (RO) process and the No-Chlorine/No-Sodium Bisulfite (SBS) Dosing process was studied. In order to prevent membrane fouling even in high recovery water operations, an advanced two-stage design was implemented to (1) control the permeate flux through the RO membrane module, (2) optimize the system to reduce contaminant build-up and (3) eliminate the use of chlorine and SBS, which can accelerate membrane fouling. The system was evaluated by monitoring the biofouling and the microorganisms proliferation on the membrane surface based on membrane biofilm formation rate (mBFR). The pilot plant was operated in the condition of a water recovery rate of 55%. As a result, the system was operated for longer than four months without membrane cleaning (clean in place; CIP) and the possibility of operation for seven months without CIP was confirmed by the extrapolation of the pressure values. In addition, the mBFR is a reliable tool for water quality assessment, based on a comparison between the fouling tendency estimated from the mBFR and the actual membrane surface condition from autopsy study and the effectiveness No-Chlorine/No-SBS Dosing process was verified from mBFR of pretreated seawater.

## 1. Introduction

In the last decade, many studies have examined ways to advance seawater desalination, focusing on: (1) Energy reduction, (2) Optimization of the seawater reverse osmosis (RO) system and (3) Desalination drainage and chemical dosing reduction to reduce marine pollution and other environmental impacts.

Kurihara et al. developed six technologies in a research project called the “Mega-ton Water System,” under a grant from the Japanese Society for the Promotion of Science (JSPS), with the aim of developing sustainable water treatment core technologies necessary for the 21st century [1,2]. And from 2016, the Saline Water Conversion Corporation (SWCC) has been conducting joint research with some members of the “Mega-ton Water System” to verify their technologies.

The technologies in the joint research were applied to achieve a reliable, high water recovery rate operation in the RO process. In seawater desalination, the water recovery rate is usually set at around 35~45% to maintain stable operation. Thus, the intake and pretreatment capacities are more than double the amount of water produced. Furthermore, several kinds of chemicals, such as chlorine, are dosed in the intake or pretreatment process. These factors increase the investment and operation costs, consume large quantities of chemicals and in addition to the operational impacts, the construction of facilities have further impact on the environment. Increasing the water recovery rate and reducing chemical dosing in the desalination system are methods for optimizing seawater RO processing and reducing the environmental impact. However, high water recovery operation with the conventional system causes the flux at the lead element to increase and sometimes it even exceeds the design criteria specified by the membrane manufacturer. Increasing the lead side element flux accelerates membrane fouling and makes the operation less reliable.

To control the lead side element flux individually, configuration should be two-staged and composed of a relatively short vessel for the first stage and a relatively long vessel for the second stage. There are two control methods. Kishizawa et al. developed a new RO system in the “Mega-ton Water System” project to increase the recovery rate with reliable operation [3]. It is based on a two-stage configuration with an inter-stage booster pump. The first stage module is operated at lower pressure in order to reduce the permeate flux of the lead element and the second stage module is operated with the pressure increased by an inter-stage booster pump. Kitamura et al. developed an advanced design RO system to increase the recovery rate with reliable operation [4,5], in their own project. They configured the same two-stage design but applied permeate back pressure instead of increasing the inter-stage pressure by booster pump, to reduce the permeate flux of the lead element. Additionally, an energy recovery device (ERD) was applied in the permeate line to exchange permeate pressure to the RO feed pressure or other energy. Furthermore, due to the ability to control the permeate flux, both designs are more suitable for applying a lower pressure RO (ultra-higher permeability) membrane compared to that in the conventional system.

Ito et al. developed a method suitable for monitoring the potential of membrane biofouling in the feedwater of RO—membrane biofilm formation rate (mBFR) [6,7]. In their study, they clarified the negative impact of continuous chlorination on RO biofouling by parallel operation of Ultra Filtration (UF)-RO pilot plants with and without chlorination. Also, they demonstrated that the RO biofouling risk can be quantified by the mBFR value. However, the testing was conducted in Muroto city in Japan and there was no evidence that no-chlorination RO operation could also be applicable to the seawater of the Arabian Gulf of the Kingdom of Saudi Arabia, which is considered to be one of the most severe seawaters for the RO process. Also, mBFR monitoring technology for the RO operating conditions in the Arabian Gulf plant will be effective.

In this study, a permeate back pressure applying (advanced design) system for high water recovery operation, No-Chlorine/No-SBS Dosing process and a biofouling potential monitoring method (mBFR) was designed and the reliability was verified in a pilot test plant in the Arabian Gulf, Saudi Arabia, which is one of the most severe biofouling areas.

## 2. Materials and Methods

### 2.1. Description of Pilot Plant

The pilot test was conducted beside the Al-Jubail Desalination Plant, in the Arabian Gulf. The seawater of Al-Jubail has higher Total Dissolved Solids (TDS) and water temperature than the seawater in other areas. Thus, a high risk of membrane fouling is assumed. The membrane cleaning (Clean in place; CIP) interval based on differential pressure increase was 1.5 months as a past record for a spiral RO membrane [8].

The seawater was fed into pretreatment, involving dual media sand filtration (DMF), then fed into the RO system. Figure 1 shows system schematics of the pilot plant and Figure 2 shows the appearance of the pilot plant.

An advanced design system developed Kitamura et al. [4,5] which is mentioned in introduction, was applied to the pilot plant. Figure 3 shows a flow diagram of the RO system.

This system is configured as a two-stage system and back pressure is applied to the permeate water of the first stage. There are fewer elements in the first stage than in the second stage, to control the permeate flux only from the lead side elements. In addition, an energy recovery device (ERD) is installed to exchange the back pressure to RO feed pressure or other energy (Permeate ERD). Back pressure was mainly provided by Permeate ERD and fine-tuned by a flow control valve. Another ERD also installed into brine line to exchange brine pressure with part of feed water of RO (Brine ERD). The pressure-exchanged feed water was boosted by the booster pump and combined with the remaining feed water boosted by the high pressure pump to supply the RO module. The flow rate of the RO system was controlled by the Variable-Frequency Drive (VFD) installed in the high pressure pump and booster pump and the flow control valves installed in the permeate line and the low pressure outlet side of the brine ERD.

Table 1 shows technical specification of pilot plant.

In this pilot plant, TM820V-400 (made by Toray Industries, Inc., Tokyo, Japan) was used as the RO element for both stages. A pressure exchanger (PX, made by energy recovery, Inc. (ERI), San Leandro, CA, USA) was used for the Brine ERD and a Pelton turbine (made by Grundfos, Bjerringbro, Denmark) was used for the Permeate ERD, which is able to prevent RO Permeate coming into contact with RO feed water.

Kitamura et al. found a relationship between the feed flowrate and foulant accumulation and optimized the configuration to minimize membrane fouling [4,5]. A smaller feed flow rate of the lead element is considered to incur a lower foulant load (foulant concentration × flow rate) on the membrane surface. On the other hand, with a smaller feed flow rate, the lower membrane surface shear force and the adhesion of foulants may increase. So, an optimum feed flowrate is important for reducing membrane fouling. In advanced design system, the number of membranes in the first stage RO is smaller. Therefore, it is easier to design module configuration with an optimum feedwater flow rate compared to that of the conventional system. By combining with the lead element flux control effect, it is possible to further reduce the fouling.

### 2.2. Operating Conditions

Table 2 shows specified operating conditions as defined by the authors of the RO process.

The RO system was operated at 55% RO recovery rate. To prevent scale formation, sulfuric acid and a scale inhibitor were dosed. Continuous hypochlorous acid and SBS dosing was not carried out in order to prevent accelerated biofouling. Sulfuric acid shock dosing (target pH: 3) was carried out once a day for 1 h to inhibit biofouling.

### 2.3. Evaluation Methods

Normalized permeate flow (converted at 25 °C) and module pressure drop (converted at 25 °C: ΔP_25_) were used as indicators of membrane performance and fouling. In general, normalized permeate flow is decreased and the module pressure drop is increased by membrane fouling [9].

Table 3 shows the criteria for CIP recommended by Toray Industries, Inc. [9], to estimate the number of consecutive days from ΔP_25_ and the change in the normalized permeate flow over time to CIP (membrane cleaning interval). The settled target membrane cleaning interval for this study is more than 3 months. This target was defined with reference to the membrane cleaning interval which is considered to be common in conventional systems with a 35~45% recovery rate [10].

In order to compare the ΔP_25_ with the conventional system, a small flat sheet RO membrane device (flat sheet membrane cell) was operated in parallel with the RO of the pilot plant under operating conditions equivalent to those of the conventional system and advanced design system. The appearance of the flat sheet membrane cell is shown in Figure 4 and the flow of the flat sheet membrane cell is shown in Figure 5.

The Flat sheet membrane cell is a compact water flow device that simulates a spiral RO membrane. The Flat Sheet RO Membrane Cell uses the same membranes and spacers as commercially available spiral wound RO membrane elements and by using equivalent flow paths and membrane surfaces, it is believed that it is possible to replicate the phenomena within RO membrane elements. Equivalent devices have been used in many other bench scale studies [11].

Biofouling and particle fouling have the characteristic of occurring in order from the leading side of the membrane. By equating the flow rate per channel area of the pilot system with that of a flat sheet membrane cell, fouling of the leading membrane can be simulated. Since the flat sheet membrane cells simulate the environment at the leading side of the membrane, the fouled part of the membrane is larger than that of the RO module, where the membrane elements are arranged in series. Therefore, the criteria for CIP shown in Table 3 do not apply to flat sheet membrane cells, because they have a larger differential pressure increase compared to the initial value than RO modules.

### 2.4. Biofouling Monitoring

The membrane biofilm formation rate (mBFR) was established to provide a simple and reliable biofouling potential quantification tool that can easily be conducted in a RO plant, for operation diagnosis and optimization [6,7]. Its container is composed of consecutively connected separable opaque plastic short columns and a RO membrane cut piece is installed in each short column as a biofilm formation base to prepare the exact same physicochemical property and the roughness of the surface of the RO element within the desalination plant for accurate biofilm development monitoring. The necessary number of unit columns can be sampled periodically or added to the existing column by twisting. Biomass that develops on a RO membrane cut piece is wiped manually with a sanitized cotton swab for a high and sure recovery of biofilm on the RO membrane surface. The biofilm is quantified by measuring the amount of adenosine triphosphate (ATP). The collected biofilm is dispersed in a tube with sanitized water and its ATP concentration is measured using the portable analysis device. mBFR is measured as pictograms of ATP per square centimeter per day as shown in Figure 6.

A guideline on the RO chemical cleaning frequency for RO plants was established based on the accumulation of mBFR data of RO feedwater together with RO operation data. From our experience, a higher mBFR value always results in earlier biofouling (increase in pressure drop). A mBFR value below 10 pg-ATP/cm^2^/d seems to be an appropriate target of RO feedwater quality to assist plant operators in preventing biofouling.

The values of mBFR in the pilot plant were collected to reflect the biofouling. The evaluation points of mBFR in the pilot testing and their purpose are shown in Figure 7.

## 3. Results and Discussion

### 3.1. Results of Pilot Plant Operation

#### 3.1.1. Water Quality and Operation History

Figure 8 shows the DMF treated water quality and the operation history of the pilot plant.

Continuous operation was started in January 2017 and continued for 186 days. The operating conditions were changed from the specified conditions as shown in Table 2 due to mechanical equipment maintenance. During this period, the feed water flowrate, which is considered to be a particularly important factor for membrane fouling, was maintained at the same level as before the change (in the range of 5~7 m^3^/h/vessel). In past operation in the same area [8], where the CIP interval based on differential pressure rise was 1.5 months, the feed flow was about 9.0 m^3^/h/vessel. Compared with this value, the feed flowrate of the pilot plant is low.

On the 81st day of operation, part of 8” RO element (2 of 38 Nos.) picks up was conducted to check the surface conditions. And on the 126th day of operation, after the Ramadan vacation, the CIP was conducted.

Table 4 shows the common quality of the DMF treated water and RO feed during operation.

Although the average of the DMF treated water TDS was 43,800 mg/L, it was 45,484 mg/L for RO feedwater due to mixing of the brine water from brine ERD. The raw water temperature during the operation period was from 14.0 to 37.9 °C and the minimum water temperature was recorded on the 15th day of operation (February 4th) and the maximum water temperature was recorded on the 148th day of operation (August 13th). The temperature of the RO feedwater temperature changed from 16.2 to 39.3 °C, which was around 2 °C higher than the DMF treated water temperature due to waste heat from the high pressure pump. SDI for RO feed was maintained below the membrane manufacturer’s operation limit of 5.

#### 3.1.2. Changes in Operation Performance

Figure 9 shows the trend of changes in feed pressure, permeate flowrate, permeate TDS and temperature. The plots from the 127th day to the 170th day of operation, which were operated under conditions different from the specified, was excluded from the graphs.

The feed pressure and the permeate TDS at the initial of operation were equivalent to those estimated from the RO projection software. Thereafter, the feed pressure remained around 8 MPa and the permeated TDS remained within the range estimated from the increase in the water temperature.

Figure 10 shows the trend of changes in normalized permeate flow and ΔP_25_. The plots from the 127th day to the 170th day of operation, which were operated under conditions different from the specified, was excluded from the graphs. And for normalized permeate flow, the plots from 0 day to the 14th day were also excluded to eliminate the initial fluctuations that occurred.

The normalized permeate flow and ΔP_25_ are indicators of membrane fouling, the normalized permeate flow decreases mainly due to fouling on the membrane surface and ΔP_25_ increases due to feed spacer clogging. Although CIP was conducted on the 126th day of operation at regular intervals, the normalized permeate flow and ΔP_25_ did not reach the CIP criteria value throughout the test period. Thus, it was verified that the advanced design system maintained stable continuous operation for at least 3 months.

The changing rate of normalized permeate flow and ΔP_25_ in this test duration was almost constant, even though the temperature increased from 14 to 30 °C. Thus, it was supposed that in the highest temperature duration the rate of change would be constant if the system continued to operate without CIP on the 126th day. The maximum continuous operation days (maximum CIP interval) were supposed to be from 200 to 210 days (6.6 to 7.0 months) based on a linear extrapolation.

#### 3.1.3. Membrane Surface Conditions (Membrane Autopsy)

Figure 11 shows the membrane surface conditions of the RO element collected on the 81st day of operation (near the target CIP interval value of 90 days) and Table 5 shows the amount of surface deposits and their composition (ash content: The ratio of the amount of ignition residues to the total amount of deposits). Only two RO Elements were collected, so as not to have a large influence on continuous operation. The collected RO elements were the lead element of the first stage and the last element of the second stage, respectively.

The averaged deposit amount of the membrane that reached the CIP criteria was around 100 g per element. But from the selected membrane, although slimy deposits were confirmed, the dry weight of deposits on the lead element were as low as 26.7 g in this test. Furthermore, the ash content from the selected membrane was as high as 51.9%, which means there was little biofilm and organic matter, although the ash content was approximately 20% when biofouling occurred. From these results, it was found that the biofouling was suppressed significantly, even in Al-Jubail where biofouling readily occurs due to the high water temperature and high organic matter content. In the last element of the second stage, no deposits were observed visually and the dry weight of the deposit was as low as 11.0 g.

#### 3.1.4. Comparison with Conventional System Using Flat Sheet Membrane Cell

Figure 12 shows the trend of changes in ΔP_25_ of the flat sheet membrane cells that were operated in parallel with the RO of pilot plant. Plots from day 13 to day 82, where the pressure was not measured correctly, were excluded from the graph.

Comparing an equivalent series to that of the conventional system, the series with an advanced design system had a lower increase of ΔP_25_. The number of days to reach 1.6 kPa was approximately 82 days for the conventional equivalent series, compared with approximately 116 days for the Advanced Design equivalent series. Therefore, it was verified that the CIP interval of the advanced design system was almost 1.4 times as long as that of the conventional system under the same pretreatment and chemical dosing conditions.

### 3.2. Results of Biofouling Monitoring

The results of biofilm development on the cartridge of mBFR is shown in Figure 13. And The values of mBFR calculated from the obtained daily increase in ATP are shown in Table 6, as water quality from the viewpoint of biofouling.

The mBFR values of raw seawater changed according to the season. The maximum values observed in April and August were about 3 times higher than the minimum value observed in February. This trend can be attributed to changes in biological activity due to water temperature and similar trends have been reported in previous studies by Ito et al. [7]. The mBFR values of DMF treated seawater remained below the target value of 10 pg-ATP/cm^2^/d. The results indicate that the DMF pretreatment of the pilot plant was effective in reducing the biofouling risk of Arabian Gulf non-chlorinated seawater, including that in the highest temperature season. For the mBFR value range of DMF treated seawater, the CIP interval is estimated to be approximately every six months according to Ito et al. [7]. In this study, H_2_SO_4_ shock dosing was used to prevent biofouling but the obtained mBFR values of DMF treated seawater suggest that the system could have been operated reliably enough without H_2_SO_4_ shock dosing.

In particular, the mBFR installed in brine can be used not only for “water quality assessment” but also “RO membrane monitoring,” since the column is installed with the same membrane used in the module. The value of the mBFR in the DMF treated water was 2 to 6 and the brine was <1, indicating that there was less or no biofilm formation on the RO membrane surface of the column. Therefore, less or no biofilm formation is expected on the RO membrane of the pilot RO module between of them, which is in agreement with the results of the membrane autopsy. This result supports the existing papers [6,7] claiming the reliability of mBFR monitoring technology for RO plant monitoring.

## 4. Conclusions

The following four conclusions are based on the results of this study:The advanced design system with No-Chlorine/No-SBS Dosing process can operate for more than 4 months and perhaps up to 7 months without any CIP, even in an area where there is high potential for membrane fouling in 55% recovery rate in the RO process.The CIP interval of the advanced design system is almost 1.4 times as long as that of the conventional system under the same pretreatment and chemical dosing conditions.The expected membrane surface condition from biofouling monitoring of mBFR in RO brine and the actual surface condition from membrane autopsy were almost the same. Thus, the mBFR monitoring technology is reliable for RO plant monitoring also in the Arabian Gulf.From biofouling monitoring of mBFR, the water quality of DMF treated seawater from the viewpoint of biofouling was preferred even in summer. Thus, the No-Chlorine/No-SBS Dosing process for reducing the potential of biofouling is effective also in the Arabian Gulf.

Thus, an advanced design system with a No-Chlorine/No-SBS Dosing process optimizes the seawater RO process and reduces the environmental impact. The results of this study will be referred to in the 10,000 m^3^/d scale demonstration project in which SWCC and New Energy and Industrial Technology Development Organization (NEDO) are collaborating.

## Figures and Tables

**Figure 1 membranes-11-00141-f001:**
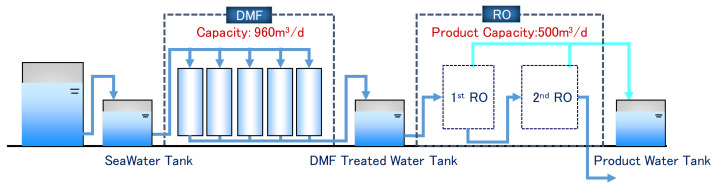
System schematics of the pilot plant.

**Figure 2 membranes-11-00141-f002:**
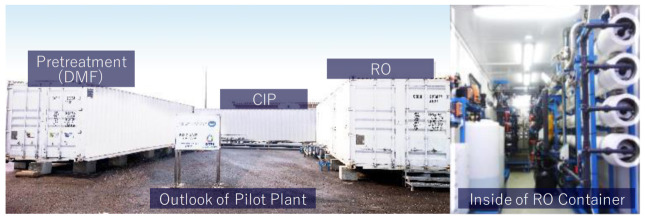
Appearance of the pilot plant.

**Figure 3 membranes-11-00141-f003:**
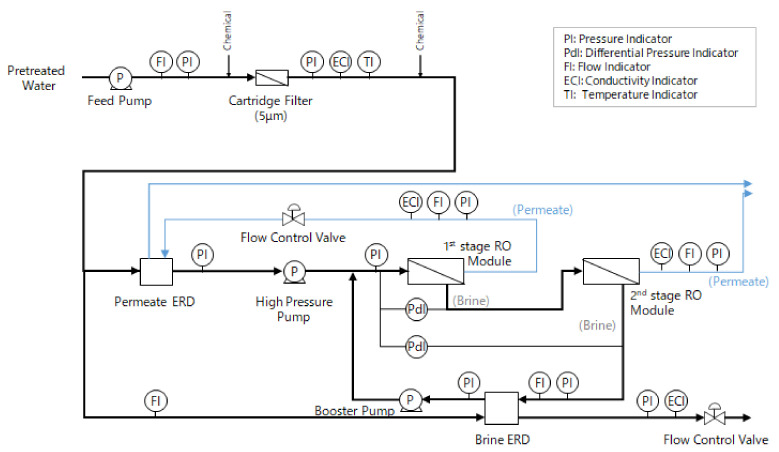
Flow diagram of reverse osmosis (RO) system.

**Figure 4 membranes-11-00141-f004:**
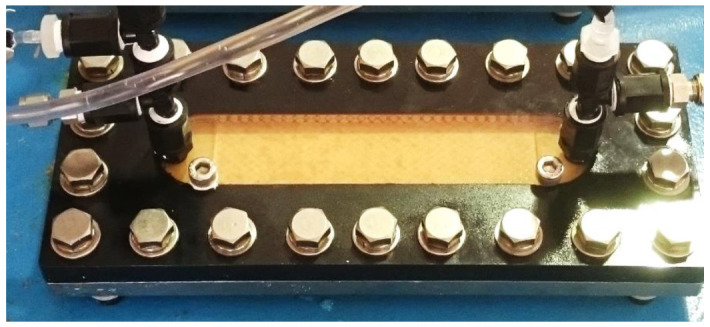
Appearance of the flat sheet membrane cell.

**Figure 5 membranes-11-00141-f005:**
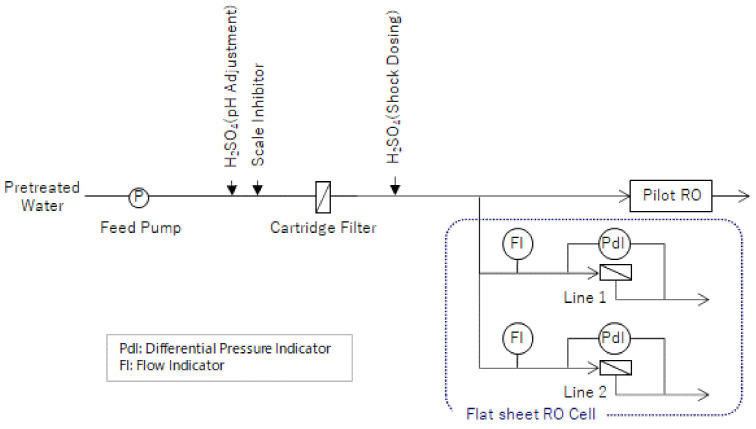
Flow of the flat sheet membrane cell.

**Figure 6 membranes-11-00141-f006:**
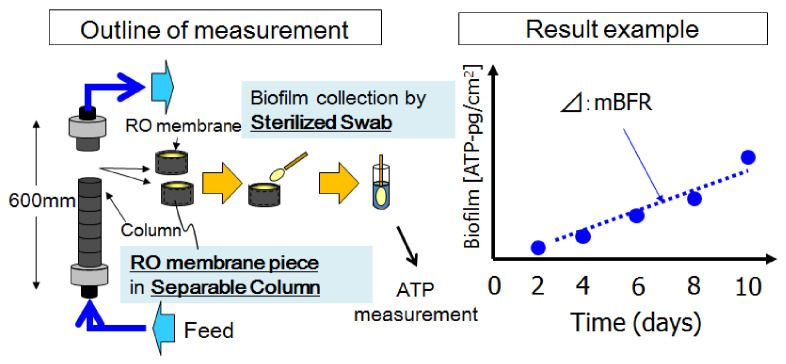
Biofouling monitoring index.

**Figure 7 membranes-11-00141-f007:**
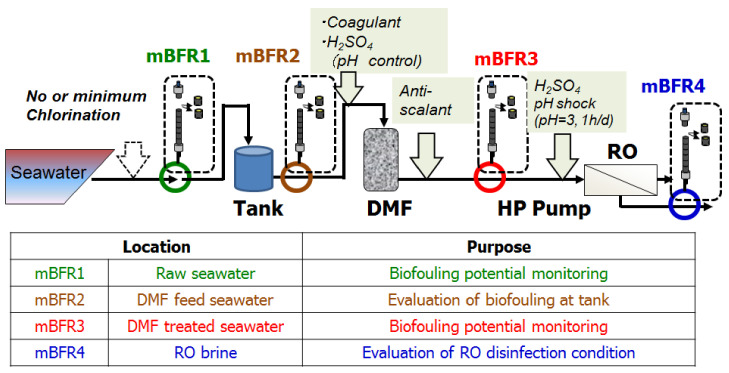
Evaluation points of membrane biofilm formation rate (mBFR) in the pilot testing.

**Figure 8 membranes-11-00141-f008:**
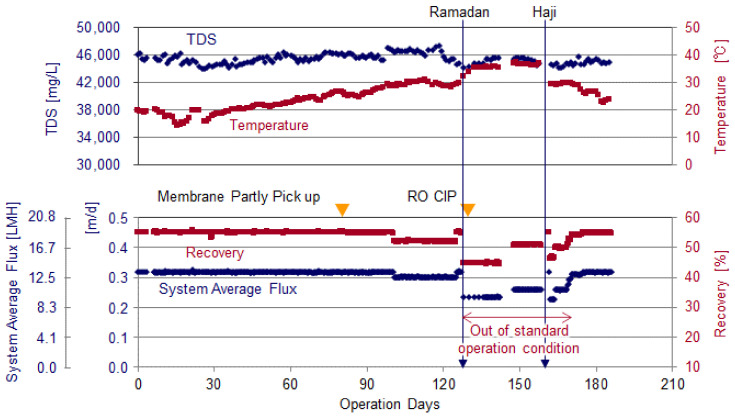
Dual media sand filtration (DMF) treated water quality and operation history.

**Figure 9 membranes-11-00141-f009:**
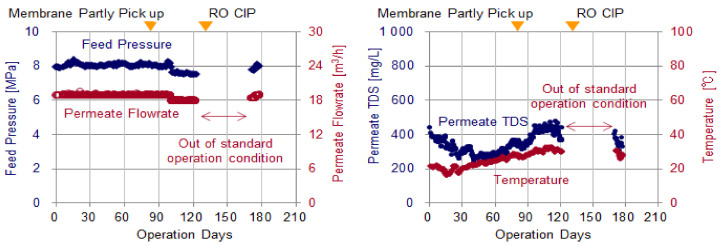
Changes in feed pressure, permeate flowrate, Total Dissolved Solids (TDS) and temperature.

**Figure 10 membranes-11-00141-f010:**
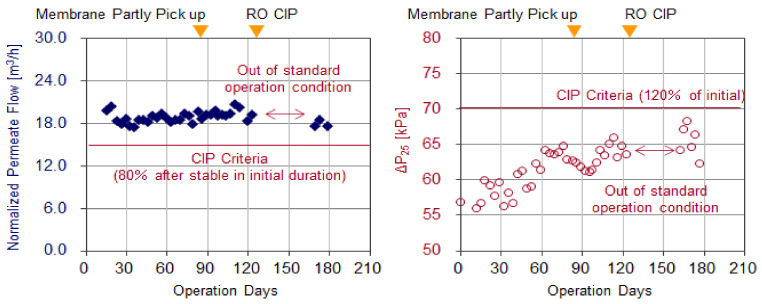
Changes in the normalized permeate flow rate and ΔP_25_.

**Figure 11 membranes-11-00141-f011:**
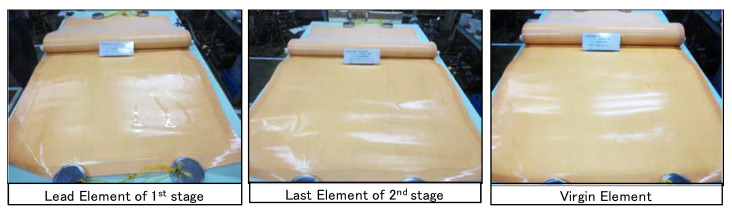
Surface conditions of the lead and last RO element (collection on 81st day of operation).

**Figure 12 membranes-11-00141-f012:**
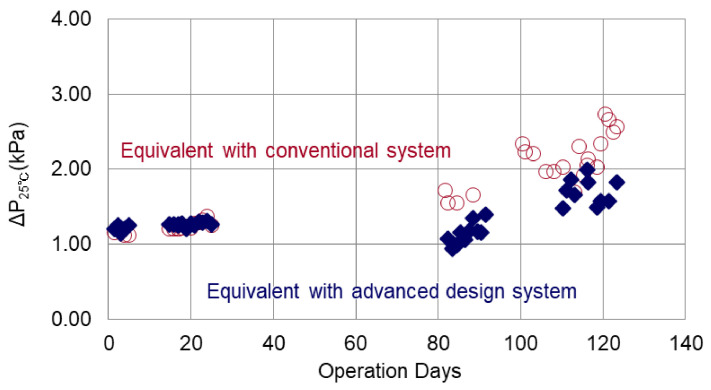
Changes in ΔP_25_ of the flat sheet membrane cells.

**Figure 13 membranes-11-00141-f013:**
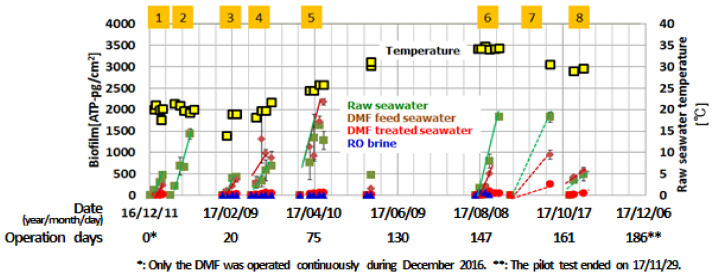
Results of Biofilm development.

**Table 1 membranes-11-00141-t001:** Technical specification of pilot plant.

Specification of DMF	
**Item**	**Specification**
Product Capacity	Average: 910 m^3^/d, Max: 1250 m^3^/d
Liner Velocity (LV)	8.7 m/h
Coagulants	FeCl_3_: 0.9 mg-Fe/L
**Specification of RO**	
**Item**	**Specification**
Product Capacity	450 to 500 m^3^/d
Membrane	TM820V-400 (37 m^2^, by Toray Industries, Inc.) (product flow: 34.1 m^3^/d, NaCl rejection: 99.8%) *
Brine ERD	Pressure Exchanger (PX)
Permeate ERD	Pelton Turbine

* at 5.5 MPa of feed pressure, 8% of water recovery 32,000 mg/L of NaCl as feed water.

**Table 2 membranes-11-00141-t002:** Operating conditions.

Major Conditions	
**Item**	**Specified Conditions**
Water Recovery Rate in RO Process	55%
System Average Flux	0.32 m/d (13.3 LMH)
**Chemical Dosing Conditions**	
**Item**	**Specified Conditions**
Chlorine	None
FeCl_3_ (for DMF)	0.6~1.2 mg-Fe/L
37% H_2_SO_4_ (pH adjustment)	pH = 7 (24 h)
Scale inhibitor	Flocon135 1.4 mg/L (24 h)
SBS	None
37% H_2_SO_4_ (Shock dosing)	pH = 3 (1 h/day)

SBS: Sodium Bisulfate.

**Table 3 membranes-11-00141-t003:** Criteria of clean in place (CIP).

Item	Criteria
ΔP_25_	120% of initial
Normalized Permeate Flow	80% after stable initial duration

**Table 4 membranes-11-00141-t004:** Common quality of DMF treated water and RO feed.

Item		MAX.	Min.	Ave.	*n*
DMF Treated	TDS [mg/L]	46,574	42,559	43,800	490
	Temperature [°C]	37.9	14.0	25.6	490
	Turbidity [NTU]	0.19	0.03	0.09	18
	TOC [mg/L]	2.2	1.4	1.8	13
RO Feed	TDS [mg/L]	48,034	42,559	45,484	490
	Temperature [°C]	39.3	16.2	27.3	490
	SDI [-]	3.90	1.50	2.87	161

TOC: Total Organic carbon, SDI: Silt Density Index, *n*: number of data.

**Table 5 membranes-11-00141-t005:** Membrane surface deposit condition and compositions.

	Lead Element	Last Element	(Example of Reached CIP Criteria) ***
Dry * Weight of Deposit (per element)	26.7 g	11.0 g	>100 g
Ash Content **	51.9%	76.1%	-

*: Dried at 108 °C, 24 h. **: The ratio of the amount of ignition residues to the total amount of deposits. ***: From experience.

**Table 6 membranes-11-00141-t006:** Results of mBFR evaluation.

	1	2	3	4	5	6	7	8
Raw seawater	69	103	54	43	121	119	(63) *	(66) *
DMF feed seawater	59	-	37	39	119	(64) *	(33) *	(78) *
DMF treated seawater	6	-	4	4	4	2	(9) *	(7) *
RO brine	-	-	<1	<1	<1	<1	-	-

* Less reliable due to insufficient biofilm data.

## Data Availability

Not applicable.

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
