# Peer review of "Reliable Sea Water Ro Operation with High Water Recovery and No-Chlorine/No-Sbs Dosing in Arabian Gulf, Saudi Arabia"

_membranes, 2021, doi:10.3390/membranes11020141_

Round 1
Reviewer 1 Report
The work is very interesting and the four conclusions are adequately justified, that is, the design optimizes the seawater RO process and reduces the environmental impact.
However, no further information is given regarding the automatic control system with which the plant is surely controlled in closed loop, for a more complete study, this reviewer recommends including this stage.
Some recommendations to improve the presentation of this paper:
- It will be beneficial for the reader if more information is added about the test pilot plant, regarding its technical specifications, sensors, actuators, other instrumentation, etc. This includes improving the scheme in figure 1.
- The paragraph between lines 108 -113 is redundant and does not add much more information to what is presented in the introduction. It is recommended to reformulate.
- In line 123, it is recommended to indicate that TM820V-400 is a type of membrane (supplied by Toray Industries) and mention its specifications in relation to energy consumption, rejection, etc.
- Verify that the title of figure 14 is on the same page as the figure.
Author Response
Thank you for your kindly advise.
The manuscript has been revised to reflect the points raised.
I would appreciate if you could give me further review.

Reviewer 2 Report
The authors present a common and always interesting problem in desalination plants using Reverse Osmosis.
The paper should be rewritten, because there are many conditions described in the paper that are not clear.
A description of points of interest is given below
ABSTRACT.
The authors should indicate what is the process used to reduce biofouling instead of say that they do not use chlorine and sodium bisulfite.
The authors say “The pilot plant implemented and advanced design”. It should be” An advanced design was implemented in the pilot plant”
The authors say “monitoring the biofouling and proliferation on the membrane surface”. It should say monitoring the biofouling and microorganisms proliferation on the membrane surface
The authors say “CIP (Clean in Place), when first the concept is described, and after the acronym is added Clean in place (CIP), also for mBFR (Membrane biofilm membrane rate)
Instead of using the term MATCHED for the CIP, the authors should say that the mBFR calculation is a reliable tool to predict the use of CIP.
INTRODUCTION
The authors say “The demonstration project applied a No-Chlorine/No-SBS (Sodium Bisulfite) Dosing process”. It is not clear if the demonstration project is the authors project or the Kishizawa et al.or Kitamura’s project. Also it should be rewritten as “a No-Chlorine/No-SBS (Sodium Bisulfite) Dosing process was applied in the demonstration project”
MATERIALS AND METHODS. Description of the plant
The authors indicate a “previous” record of 1.5 months clean in place interval, but they do not indicate if this is a regular schedule.
The authors give a general diagram in Figure 1, showing a two stage RO plant, but in Figure 3 the authors give a more detailed diagram. In This Figure 3 it is not clear what is the energy exchanged in the Brine. Readers could imply that there is only heat exchange, presumably for the permeate EDR, but the authors should indicate if there is also a mechanical energy recovery in the brine ERD.
Also the “box” used for the permeate ERD should be similar to the one used for the brine regardless of the nature of the energy exchanged
Also it is , at least, interesting the use of a booster pump instead of another high pressure pump unless there is mechanical energy exchange.
It is also confusing that the authors say that they use the design of Kitamura as they mention in the introduction, “Kitamura et al. developed an advanced design RO system to increase the recovery rate with reliable operation [4][5], in their own project. They configured the same two-stage design but applied permeate back pressure instead of increasing the inter-stage pressure by booster pump, to reduce the permeate flux of the lead element.”, and in Figure 3 there is a Booster pump that seems to be inter stages.
It is not clear why part of the feed has to exchange energy with the brine of the second stage and then boosted to high pressure and fed to the first stage.
The authors indicate that “There are fewer elements in the first stage than in the second stage, to control the permeate flux only from the lead side elements.” It would be more interesting for the authors to indicate the total area of the installation and what is the disposition of the modules in both stages.
The authors indicate that “TM820V-400 was applied as the RO element for both stages” may be “used” instead of “applied” is more appropriate
It is not clear the use of operating conditions shown in table 1. Do these conditions are a calculation base to evaluate economy? Do these conditions serve as point of comparison? Table 1 indicates 50000 m3/d when in this paper the objective is 500 m3/d. Why in these conditions pressure applied is not considered?
The authors indicate in Fig 4. A “Conventional system” that is not clearly defined
Moreover, in Figure 4 there is a comparison between high rejection membranes with low energy membranes and conventional system and advanced design system. Several questions arise:
What is a conventional system? Is the area of the different systems the same?
Can the high rejection membranes be low energy membranes and vice versa?
Why one has to compare high pressure with low energy and cannot be high pressure with low pressure and compare high energy with low energy?
What is the definition of low energy membranes?
Is Figure 4 a result of this paper or is Figure 4 a result of a previous study, and if so, why no reference is given?
Are the feed flow conditions and pressures the same for all conventional and advanced?
What is the maximum lead element flux? Why it is considered maximum? Is the maximum flux referred to the flux without backpressure? The effect of back pressure is usually related to the reduction of permeate, and related to the water recovery per element. It also reduces effective applied pressure.
Is the lead element the first module of the first stage? And why there is only one module? Why the disposition and number of elements has not been indicated? According to the Manufacturer specifications, one TM820V-400 module has 37 m2 with a rejection of 99.8% and a feed spacer of 34 mil (is this considered a low energy module with respect to the TM820V-440 that has a 28 mil spacer, and 41 m2?)
The authors indicate “It was also confirmed that the SEC and the operating cost were minimized at around 45% and 53% of the water recovery, respectively” and then they indicate they are going to use 55% recovery, which makes no sense. The authors should simply say they want a 55% recovery and then find how the specific energy consumption and the operating costs would be affected. Even more, since it is is likely that the area of the systems is not the same, a total cost evaluation would be even more interesting.
The authors indicate that feed flowrate is important according to Kitamura results. It would be also interesting to compare those values with the ones of the manufacturer data sheet. It seems that for a 34.1 m3/d at a recovery rate of 8% feed flow rate would be 426.3 m3/d per module, and that would be the recommended feed flowrate producing 0.10 MPa of pressure drop per element
MATERIALS AND METHODS. Operating conditions
The authors indicate an average permeation rate of 0.32 m/d. If 500 m3/d are being produced, a membrane area of 1562 m2 would be required or 42 membrane modules, again, no distribution is given for the two stages.
The authors should clarify if normalized permeate flow is also referred to 25ºC
The authors settle a target for the CIP interval in three months, but there is no reason behind. What is the CIP interval for a 35% recovery in a conventional system?
The authors use two small flat membrane modules (no area is given, but it would seem of square centimeters looking at Figure 5) to make a comparison of what happens to a 1500 m2 installation. It seems quite a difficult comparison.
RESULTS AND DISCUSSION. Results of pilot plan operation. Water quality and operation history
Fig 9 shows the evolution of four parameters with time beginning in January 2017. What seems not relevant for the biofouling is the reference to Religious beliefs of the authors. Especially, because those events change date year after year and are not a key reference to the process.
The authors indicate the conditions were changed from “standard” conditions. It is not clear what standard means for the authors, maybe the conditions were changed from the initial established conditions due to maintenance purposes
Authors include in Table 4 a column with a parameter “n” that is not described. On the other hand, Raw water has a silt index below the maximum recommended by the manufacturer that is 5
Authors at this point reveal that the ERD for the brine is actually a mechanical energy exchange due to the mixture of the final brine and part of the feed.
RESULTS AND DISCUSSION. Results of pilot plan operation. Changes in Operation performance
Figure 10. shows that the authors are running the membrane modules close to the maximum operating pressure recommended by the manufacturer. It seems that a deviation from such pressure causes a significant quality of permeate while a decrease in membrane rejection from 99.3 down to 98.5%. Temperature increase also affects rejection, but in a much lower extent from 99.3 down to 99.1%. It is advisable to remove Religious references and ask the authors respectfully to keep them in their private environment.
Figure 11 is The best of the figures of the paper, but the scale of the DP25 should be from 50 to 70 kPa to show the readers a better view of the fouling occurring at the membrane surface
RESULTS AND DISCUSSION. Results of pilot plan operation. Comparison with Conventional System Using Flat Sheet Membrane Cell
At this point, the authors compare the advanced design system with the “conventional “ system represented by the small RO flat membrane units. As a main critical point, the authors could have saved money testing and advanced design system in flat membranes also.
Fig 12 does not show the same trend of DP25 as Figure 11 shows, in fact the values are 1/10 of those of Figure 11. Even more, absolute values should start from the same value, and there is already 0.5 kPa of absolute difference
In this case opposite to Figure 11, the authors have not included the same maximum DP25 criterium. If one applies 120% of the initial DP25 to the flat sheet, the maximum DP25 would be approximately 0.75 kPa, and the maximum DP25 for the advanced would be approximately 0.3 kPa. Moreover, the slope of both systems is higher for the flat membranes, but very similar to the advanced system. That is: the flat membrane slope is 0.013 DP25 per day, and the advanced system is 0.0083 0.013 DP25 per day, which is not exactly half as the authors claim, but 62%
RESULTS AND DISCUSSION. Results of pilot plan operation. Results of Biofouling Monitoring
To begin with, the dates of the abscissa of figure 13 should be announced as year/month/day, otherwise there is a little confusion to the reader. But more important is the effect of raw water cleaning and chemical addition. In fact, results of figure 13 show that the main anti biofouling claimed by the authors which is the shock using 37% sulfuric acid is not so crucial, since before that shock treatment the values of ATP for biofilm formation are negligible, and by the way, not readable in Figure 13.
It is not clear if values in Figure 13 are the values given in Table 6 and the evolution of values given in Table 6.
Author Response

(The authors gave the same response as above.)

Reviewer 3 Report
The Authors presented the research desalination of the sea water with high water recovery and no-chlorine /no-SBS dosing process and additionally the Authors was evaluated system by monitoring the biofouling on the membrane surface based on the membrane biofilm formation rate (mBRF). The work can be a valuable collection of information and guidance for seawater treatment technologists. The results are clearly explained and discussed with other authors and the tables are transparent and the figures are clearly. Overall, I found the results are very interesting and convincing. I recommend this manuscript to publication after minor cerrections.
I suggest completing the details of the RO membrane used in the pilot installation (membrane dimensions, active surface, membrane material, manufacturer). On the page 7, line 225 should be …….Figure 8.
Author Response

(The authors gave the same response as above.)

Reviewer 4 Report
The author investigated the effectivity of the combination of the improvement of 22
water recovery ratio in the RO process and the No-Chlorine/No-SBS Dosing process on the pilot desalination plant. The author clearly demonstrated the proposed system can operate for longer time without cleaning. I would recommend it for acceptance after the minor points listed as follows are addressed.
Page 11 Figure 14
The operation days also should be put down with date.
Page 12 Line 337
Why did the mBFR values of raw water change with the season? How was the relationship between the biofouling and temperature? Did the author characterized the effect of the biofouling on the membrane operation parameter (such as dP and flux)?
Page 12 Line 348
The author described that the mBFR results agree with the results of the RO operation and RO element autopsy. Although the surface composition results indicate the low biofouling, it is not clear the mBFR correctly reflect and monitor the membrane fouling.
Page 12 Line 360 (conclusion 3)
Which results did the author lead this conclusion from? The author didn't show the expected CIP interval from mBFR, although expected it from the actual experiments.
Author Response

(The authors gave the same response as above.)

Round 2
Reviewer 2 Report
The revised version of the manuscript has shown a significant improvement. Some issues may still need to be addressed. Detailed comments follow:
The reference to Acronyms was changed as suggested, but they have included extra acronyms without definition, such as LV and VFDs and ERI’s this latest being a company (Energy Recovery Incorporated)
It is not clear if Figure 3 includes both diagrams or only the new one is shown. The text control of changes used by the authors introduces more confusion and difficulty of reading. It also applies to former Figure 4
The authors insist in using the term standard conditions to their operating conditions. If those are a standard please give a reference if this comes from a certified institution, such as ASME or IEEE, or another
Operating conditions is more than enough and do not lead to confusion
Normalized permeate flow only requires a reference temperature to be able to compare mass balances. It is not required feed concentration, pressure applied, permeate back pressure or recovery rate.
The authors believe that the flat cells are comparable units for the biofouling measurement. This reviewer considers that it would have been more appropriate to use one of the Toray modules for that purpose. It would be a comparison of two elements of the same size and conditions that the “advanced system”. Otherwise, the authors could have used a flat cell in the “advanced system configuration”.
The authors keep misusing the term standard to what is it is believed refers to simply the chosen operating conditions
The authors, instead of indicate the feed flowrate discuss that 426.3 m3/d as feed flow is too high and thar around 240 m3/d would be more appropriate. That is very interesting, but does not clarify what the authors use. In fact 426.3 m3/d is the result of using the Toray’s data from the specification sheet https://www.toraywater.com/products/ro/pdf/TM800V.pdf
Using 5.52 MPa, 32000 mg/L of NaCl, 25ºC and 8% recovery producing 34.1 m3/d of permeate at 99.8 % rejection. If 34.1 is the 8% of the feed, then 34.1/0.08 =426.3 m3/d
It maybe high, or low or intermediate, but is the value calculated from the specification sheet.
In fact the MINIMUM permeate flowrate is 28.4 m3/d, that would imply 355 m3/d if an 8% recovery is assumed
The reviewer with this argument tries to say that is irrelevant to discuss if is low or high when the authors do not clarify what they use.
The more important claim in the paper that is the 37% H2 SO4 shock treatment as almost effective antifouling is not supported by the evidence shown in former figure 14, now figure 13 and table 6, identifying the sampling points indicated in former figure 8, now figure 7
According to now Figure 7 The shock treatment is performed before sampling point mBFR3, and those data are reflected in now figure 13 and in table 6. Those data indicate a negligible too low rate of fouling compared to previous sampling points mBFR1 and mBFR2, indicating that the coagulant, H2SO4 for pH =7 and the antiscalant are effective enough. It is also clear that the shock treatment reduces biological growth down to negligible, but one could also consider that brine concentration rises up 90000 mg/L and at that concentration there is no microbial growth. Have you checked?. The discussion seems more appropriate about the inclusion or the optimization of the shock conditions, why 1 h per day and not 30 min, or even 15 min
Moreover, the authors consider a sensitive information the arrangement of the modules, indicating that the shock is not the only responsible for antifouling in the system.
Maybe the authors should release a patent or commercialise the system arrangement, make a commercial publication or present a communication in a international congress rather than publish a scientific paper
Author Response

(The authors gave the same response as above.)
